# Presumed Roles of APRO Family Proteins in Cancer Invasiveness

**DOI:** 10.3390/cancers14194931

**Published:** 2022-10-08

**Authors:** Yuka Ikeda, Kurumi Taniguchi, Haruka Sawamura, Sayuri Yoshikawa, Ai Tsuji, Satoru Matsuda

**Affiliations:** Department of Food Science and Nutrition, Nara Women’s University, Kita-Uoya Nishimachi, Nara 630-8506, Japan

**Keywords:** APRO protein, microRNA, matrix metalloproteinase, tissue inhibitors of metalloproteinase, exosome

## Abstract

**Simple Summary:**

Here, we discuss the invasiveness of cancer cells in relation to APRO family proteins on the basis of understanding the function of matrix metalloproteinases (MMPs) and/or exosomes. Although APRO family proteins could regulate cancer invasiveness, alternative consequences might occur due to the distinctive effects of MMPs and/or exosomes containing certain microRNAs. Such knowledge could be of use to bring about novel strategies for cancer therapy.

**Abstract:**

The APRO family members may be involved in the regulation of cell growth, migration, and/or invasion. Although an APRO protein could suppress the invasiveness of several cancer cells, it has been reported that overexpression of the same APRO protein could also promote the invasiveness and/or metastasis of the same cancer cells. In general, the invasiveness of cancer cells might be associated with the function of matrix metalloproteinases (MMPs) as well as with the function of certain exosomes. However, it has been shown that exosomes involving particular APRO proteins, MMPs, and/or microRNA could contribute to the regulation of invasiveness. Here, we discuss contradictory reports on invasiveness in relation to APRO family proteins on the basis of understanding the function of MMPs and/or various exosomes. A better understanding of those mechanisms could be of use to bring about innovative strategies for cancer treatment.

## 1. Introduction

The APRO family is composed of at least six distinct members in vertebrates, namely, Tob1, Tob2, BTG1, BTG2/TIS21/PC3, ANA/Tob5/BTG3, and PC3b [1]. The main characteristic of this family is the presence of a highly conserved, 110-amino-acid N-terminal region, designated as the APRO homology domain [1], which holds two highly homologous regions, designated Box-A and Box-B [1,2]. Box-A has been suggested to play an antiproliferative role [2]. All the APRO family members may be involved in the regulation of cell proliferation, and actually, have the potential to regulate tumor cell growth [1,3]. Tob1 was isolated as a protein associating with the ErbB2 receptor protein [3]. Subsequently, Tob2 was isolated on the basis of its similarity to Tob1 [4]. Other family members had been identified using several different strategies [5]. A significant association has been identified between the expression level of Tob1 and clinicopathological characteristics, including the depth of invasion and/or the lymph node metastasis stage [6]. The downregulated expression of Tob1 has been found in malignant gastric cancer, suggesting that the low expression of Tob1 may be an independent indicator of poor prognosis in gastric cancers [7]. Similarly, the downregulation of Tob1 may be associated with the shorter survival of gastric cancer patients [8]. Consistently, Tob1 overexpression could not only increase the expression of PTEN, but also regulate the downstream effectors in the PI3K/PTEN signaling pathway, leading to the suppression of cancer cell proliferation [9]. In addition, significant prognostic effects of the whole APRO family have been found in lung adenocarcinoma and ovarian, colorectal, and brain cancers, but not in squamous-cell lung carcinoma [10]. Thus, accumulating evidence has shown that APRO family proteins may function as a tumor suppressor.

However, it has been also reported that Tob1 expression may be upregulated during the progression of colon cancer, which is significantly correlated with tumor size and a prognostic indicator, such as survival rate in colon cancer [11]. In addition, Tob1 deficiency appears to lead to the reduced tumorigenesis in DSS-treated cancer, suggesting that Tob1 is an adverse prognostic factor [11]. Consistently, the downregulation of Tob1 is associated with the shorter survival of gastric cancer patients [8,12]. The suppression of Tob1 could enhance the metastasis in lung carcinoma cell line A549 cells [9,13]. To summarize the above, although Tob1 could suppress the proliferation and/or carcinogenesis in a cell, Tob1 overexpression could also promote the invasiveness and/or metastasis of several cancers. The invasiveness and metastasis of tumors are associated with matrix metalloproteinases (MMPs) [14,15]. In addition, invasiveness, metastasis, and concomitant poor prognosis may be also involved in the function of certain exosomes [16,17]. Here, we discuss contradictory reports on the invasiveness of cancer cells in relation to the actions of Tob1, a member of the APRO family, on the basis of understanding the function of MMPs and/or exosomes. We wish we could combine MMPs and exosomes in the context of APRO regulation for the invasiveness of cancer cells. A better understanding of these mechanisms might be used to design more efficient cancer therapies [18]. Such knowledge could also be of use to engineer novel strategies for cancer treatment.

## 2. Exosomes with APRO Proteins and/or Certain MicroRNAs May Contribute to Cancer Invasion

Exosomes are a class of extracellular membrane vesicles with a circular lipid bilayer ranging in diameter from about 30 to 150 nm [19], which are capable of mediating invasion and/or metastasis by transferring their contents, including proteins, lipids, and nucleic acids, to adjacent cells [20]. Exosomes are broadly distributed in blood cells, dendritic cells, tumor cells, and other cells [21], which can be used for diagnosis and/or prognosis within cancer patients [22]. Molecular machineries of prevailing biogenesis, cargo loading, and/or delivery of exosomes may be intricate and are still under investigation. Exosomes secreted from gastric cancer cells overexpressing Tob1 could induce autophagy of LC3-II accumulation in the gastric cancer cells [23], which may influence the proliferation, migration, and invasion of cancer cells [24]. It has been shown that exosomes containing the BTG1 protein are present in the pleural fluid obtained from patients suffering from mesothelioma, lung cancer, breast cancer, and ovarian cancer [25]. The BTG1 protein has been also identified in plasma exosomes. In addition, the plasma-exosome-derived BTG-1 levels have been related to tumor diameter, stage, tumor metastasis, the degree of tumor differentiation, and abnormal CEA levels, in accordance with previous findings of BTG-1 expression in other cancers [25]. Furthermore, a low number of plasma exosomes with low levels of BTG-1 have been observed in the poor differentiation group, suggesting that plasma-exosome-derived BTG-1 might be a potential biomarker for a prognosis in patients with non-small-cell lung cancer [25].

Noncoding RNAs in exosomes from a variety of cells have been shown to influence metastasis via various mechanisms [26]. In particular, the microRNAs (miRNAs) commonly detected in exosomes are single-strand, non-encoding RNAs with the length of about 20 nucleotides, which are usually found in eukaryotic organisms [27]. The miRNAs can affect the stability and/or translational efficacy of their target mRNAs, consequently resulting in decreased protein translation [28]. More than 1000 miRNAs have been shown to regulate a lot of biological processes including proliferation, migration, and/or differentiation in cells [29]. Interestingly, miR590 in the exosome exerts its effects through targeting Tob1 [30]. miR32 might regulate the invasion of cancer stem cells, as it is upregulated in colorectal cancer tissues compared to the adjacent normal tissues [31,32]. An exosome containing miR-135a-5p could activate MMP-7 to promote liver metastasis in colorectal cancer [33] (Figure 1). Exosomal miR-4443 may also promote the metastasis of breast cancer cells through downregulating tissue inhibitors of metalloproteinase 2 (TIMP2) and upregulating several MMPs [34]. For enhanced migration, exosomal miR-21 may influence MMP-9 and TIMP-2 through the PI3K/AKT signal pathway, but not MMP-2 and TIMP-1 [35]. TIMP3 might be targeted by macrophage-derived exosomal miR-21-5p [36]. MMP-2 expression might negatively correlate with miR-29c expression in urinary exosomes [37]. These findings are indicative of the potential of miRNAs and/or exosomes as therapeutic markers in various cancers.

## 3. MMPs and TIMPs Could Be Also Involved in Cancer Invasion and Metastasis

MMPs are key players in matrix remodeling. Their function has been principally investigated in cancer biology; they are involved in different steps of cancer development, from local expansion by the proliferation of cancer cells to tissue invasion and/or metastasis through extracellular matrix degradation [38]. MMPs could promote cell migration and tumor invasion through the proteolytic degradation of the extracellular basement. MMPs are synthesized as pre-proMMPs, from which the signal peptide is removed during translation to produce mature proMMPs. MMP expression can be also affected by several hormones, growth factors, and/or cytokines [39]. A higher expression of MMPs has been revealed as a potential marker of higher invasiveness and/or worse prognosis in patients with various cancers. For example, ovarian hormones could affect the expression of several MMPs, which might participate in endometrial tissue remodeling during menstrual cycles [40]. Additionally, increases in estrogen and/or progesterone, as well as vascular endothelial growth factor (VEGF), during pregnancy could promote the expression of several MMPs, which might also facilitate the tissue invasion of cytotrophoblasts [41]. Exosomes from vascular smooth muscle cells (VSMCs) may be burdened with the MMP-2 protein and specific miRNAs for controlling cell adhesion and/or migration [42]. Furthermore, macrophage-derived exosomes could trigger the expression of MMP-2 in the VSMCs via JNK and p38 pathways [43]. Interestingly, the Box-A domain in the Tob1 protein may have protease activity, which activates the MMP-7 [44]. It has been discovered that MMPs exist in exosomes from various cell types and/or some body fluids [45].

The activity of MMPs could be regulated by endogenous tissue inhibitors of matrix metalloproteinases (TIMPs). In general, MMPs are regulated at multiple levels, including in mRNA expression, the activation of the proenzyme to the active form, and the counteracting actions of these TIMPs, which are specific for each MMP type. There are four homologous members in the TIMP family with a similar structure [46], which could also regulate remodeling and turnover of the extracellular matrix (ECM) during normal and/or pathological conditions [47]. The N-terminal domain of each TIMP protein holds the inhibitory activity for the wasting potential of the MMPs [48]. The role of TIMPs in the ECM turnover could be defined as the potential inhibition of MMPs with various efficacies. Increased MMP activity and/or decreased TIMP expression could lead to MMP/TIMP imbalance, which might result in various pathological conditions including cancer invasion and/or metastasis. For example, TIMP-1 has been shown to interact with several MMPs and the matrix-degrading properties of the MMPs, which could play a fundamental role in the spread of cancer [49]. In addition to the MMP-inhibitory function, TIMP-1 could also stimulate cell growth [50], and it exhibits antiapoptotic activity [51]. It has been reported that exosome-bound TIMP1 may be a circulating biomarker for a noninvasive risk stratification in patients with colorectal liver metastases [52]. In addition, it may be extremely possible to diagnose cancers by precisely analyzing the expression of TIMP-1 in exosomes [53].

## 4. Activated MMPs and/or Certain MicroRNAs in Exosomes Could Contribute to the Enhanced Migration, Invasion, and/or Metastasis of Cancer Cells

In general, various MMPs are upregulated in various cancers and inflamed regions. Cancer progression could be a complex process, during which numerous cells, including malignant cells, inflammatory cells, and/or surrounding stromal cells, might communicate with each other in the microenvironment. MMPs may be involved in the remodeling of the extracellular matrix in the microenvironment to allow dissemination and/or metastasis of cancer cells [54]. Among them, MMP-2 and MMP-9 are the most distinctive MMPs characterized by a strong proteolytic activity in the extracellular matrix [55], which could be overexpressed in tumor cells and may be linked to risky metastasis and/or poor prognosis [56] (Figure 1). Exosomes could also regulate the migration of lung cancer cells into the rich vasculature by promoting MMP-2 expression [57]. In addition, exosomes could activate MMP-2 to enhance the invasiveness required for the first step in the metastasis of cancer cells [58]. Furthermore, it has been shown that exosomes derived from renal cancer cells may contribute to renal cancer development, progression, and/or invasion via the increased expression of MMP-9 [59]. Additionally, exosomes with MMP-13 could enhance migration and/or invasiveness to promote the aggressiveness of nasopharyngeal carcinoma cells [60]. High levels of MMP-1 in exosomes could potentiate the metastasis in triple-negative breast cancer [61]. Exosomes from cancer stem cells could enhance the proliferative, migratory, and/or invasive abilities of fibroblasts, accompanied by the upregulated expression of MMP-2 and MMP-9 [62].

Exosomes could promote the proliferation, migration, invasion, and angiogenesis of HUVECs, in which the increased mRNA and protein levels of VEGF and/or MMP-9 are detected [63]. Melanoma-derived exosomes may also provide an invasive capability with the higher expressions of MMP-2 and/or MMP-9, in which miR-21 is at least partially responsible for the effect [64]. Exosomes with miR-205-5p could promote angiogenesis and metastasis by enhancing the expression of MMP-2 and/or MMP-9 [65]. Exosomal miR-106b could enhance the invasive ability of lung cancer cells and increase MMP-2 and MMP-9 expression [66]. Exosomal miR-4435 could affect the migration and/or invasion of colorectal cancer cells [67]. Interestingly, exosomes with CRN2, which is an actin filament-binding protein involved in the regulation of cell migration and invasion, could promote perivascular invasion of glioblastoma cells by increasing the catalytic activity of MMP-14 [68]. Similarly, exosomal CXCR4 could promote hepatocarcinoma cell migration, invasion, and/or lymphangiogenesis by enhancing the secretions of MMP-2 and MMP-9 [69].

MMPs might be crucial for ECM remodeling under the pathological conditions of cancers. MMPs are of crucial importance for the invasiveness of cancer cells. For a good invasion performance, cancer cells must adjust the activation rate of MMPs corresponding to the solidity of the surrounding ECM. Accordingly, their expression may correlate with metastatic potential and to a significant prognostic marker.

## 5. Activated MMPs Could Also Regulate the Responses of Immune Cells against Cancers

Cancers must evade antitumor immune responses to continue to grow. In fact, cancer cells can often escape from immune surveillance, which has been shown to be associated with various types of immune cells including Tregs and Th17 cells [70] (Figure 1). Therefore, immune responses against cancer have been revealed as a crucial issue in the treatment of cancer. Most tumor cells express antigens that can mediate recognition by host CD8+ T cells. Interestingly, high levels of MMP-9 detected in laryngeal cancer could play a critical role in the development of Treg cells, which have an ability to suppress the tumor-specific CD8+ T cells [71]. In addition, the increased production of MMP-7 might trigger an increase in the suppressive function of Treg cells [72]. Additionally, the expression of MMP-9 might be correlated with the markers of Th1 cells and/or T-cell exhaustion [73]. Furthermore, upregulated expressions of MMP-2 and MMP-9 may promote the migration and/or invasiveness of esophageal adenocarcinoma via the action of IL-17A, which is a proinflammatory cytokine secreted from Th17 cells [74]. Likewise, the MMP inhibitor may regulate the expression of TGF-β, thus reducing the number of Tregs [75]. Amazingly, the expression of MMP-7 caused by *H. pylori* infection could contribute to poor responses of the adaptive immune system characterized by insufficient Th1 and/or Th17 cells and the inappropriate activation of Treg cells [76,77]. Human chorionic gonadotropin (hCG), a hormone essential for pregnancy, is also ectopically expressed by a variety of cancers and is associated with a poor prognosis, which could induce the synthesis of MMP-2 and/or MMP-9, thereby increasing invasiveness in an MMP-dependent manner. The hCG could also upmodulate the secretion of TGFβ and IL-10, thereby inhibiting T-cell proliferation [78].

Consistently, the inhibition of MMP-2/MMP-9 may improve the efficacy of PD-1 or CTLA4 blockade therapy in the treatment of aggressive metastatic cancers [79]. The PD-1 or CTLA4 checkpoint blockade are dramatic therapies for several cancers that enhance antitumor immune activity. Immune checkpoints are diligently related to tumor immune escape, which may be related to the poor prognosis of some tumors in the survival analysis [80]. The PD-1 ligand is regulated through proteolytic cleavage by endogenous MMPs from stromal fibrocytes (Figure 1). For example, increased MMP-10 expression in CD90+ fibroblasts may contribute to mucosal tolerance via the suppression of Th1 cells through the cell surface membrane-bound PD-L1, which could suppress Th1 and/or Th17 responses from activated CD4+ T cells [81]. In this case, supplementation of the MMP inhibitors could restore the suppression of Th1/Th17 cells. PD-L1 could be also cleaved by MMP-13, whereas PD-L2 is sensitive to broader MMP activities. Accordingly, MMPs might play a significant role in the immune checkpoint responses in cancer therapy. In fact, the MMP-dependent cleavage of PD-1 ligands on fibroblasts may limit their immunosuppressive capacity [82]. Interestingly, a combined treatment with the MMP inhibitor and anti-CTLA-4 antibody could delay tumor growth and reduce the metastases compared with anti-CTLA-4 treatment alone in lung and liver cancers [83]. Similarly, MMP-9 inhibition with an anti-MMP-9 monoclonal antibody could promote antitumor immunity through the disruption of biochemical barriers to the T-cell trafficking of tumors [84].

## 6. Discussion and Perspectives

There are contradictory reports on the invasiveness of cancer cells in the exploitation of the BTG1 protein, the other member of the APRO family, as well as those of Tob1 as shown in the introduction. A body of evidence indicates that BTG1 expression is negatively correlated with tumor invasion, lymph node metastasis, the clinical stage, and/or a low survival rate in patients with various cancers. For example, the low expression of *BTG1* might be involved in the progression of pancreatic ductal adenocarcinoma, suggesting that BTG1 might be a poor prognostic marker of the survival rate in cancer [85]. *BTG1* expression has been shown at lower levels in colorectal cancer than in the control, due to the hypermethylation of the *BTG1* promoter [86]. In addition, the low expression of *BTG1* has been reported to be associated with aggressive features and/or a worse prognosis of thyroid cancer [87], esophageal cancer [88], and squamous cell skin carcinoma [89]. The cumulative survival rate of BTG1-positive patients with colon cancer is significantly higher than that of *BTG1*-negative patients [90]. However, it has also been reported that *BTG1* expression is positively linked to aggressive features of colorectal cancer, including depth of invasion and/or lymph node metastasis [91]. In addition, colorectal metastatic cancer cells in the lymph nodes has shown more *BTG1* expression than that in the primary cancer site, suggesting that the overexpression of BTG1 might promote the invasion and/or metastasis of the colorectal cancer [91,92]. Similarly, the BTG1-overexpressing endothelial cells have exhibited increased cell migration [93]. Taken together, *BTG1*’s high expression might be involved in the poor progression of several cancers, and might be considered as a marker indicating that BTG1 promotes migration, invasion, and/or metastasis. Consequently, we should be cautious to employ APRO proteins for the target of cancer therapy.

How is this situation explained? Our answer is as follows: It has been shown that the ubiquitin–proteasome system could mediate the degradation of many proliferative and antiproliferative gene products. Therefore, this system might play an important role in the degradation of APRO proteins, as well as unusual key proteins [94]. In fact, the degradation of BTG2 is inhibited by lactacystin, a proteasome-specific inhibitor [95]. This ubiquitin–proteasome system could be affected by the alteration of cellular homeostasis that is cell-type dependent [96], which may be one of the reasons (reason 1). A balance between the expression of MMPs and that of TIMPs could control cancer cell migration, invasion, and/or metastasis. Both the synthesis and degradation of these proteins are important for determining how they work. As shown here, the balance of these proteins might be cell-type or cancer-type dependent, which might be the other reason (reason 2). In opposition to the invasion-facilitating exosomes, as shown in Section 2 and 3, some exosomes could slow down angiogenesis, migration, invasion, and/or metastasis [97,98,99], which might be also cell-type or cancer-type dependent (reason 3). APRO family proteins might be a key modulator of microRNAs [100]. Even though APRO members could inhibit cell proliferation, migration, and/or invasion, the other conditions such as the expression levels of MMPs or TIMPs, presence of exosomes, and/or functions of proteins/microRNAs in exosomes could further modify the effect of APRO proteins, probably for those reasons (Figure 2). However, the pathophysiological significance of APRO family proteins in cancer remains unknown. Evolving evidence suggests that PIWI-interacting RNAs (piRNAs) may be important epigenetic regulators of gene expression in human cancers [101], which may also significantly contribute to cancer pathogenesis [102]. BTG1 has been shown as a direct target of piR-1245, suggesting an inverse correlation between BTG1 expression and piR-1245 in colorectal cancer [101]. BTG1 has also been shown as a direct target of miR-330-3p, which could increase the expression of MMP-9 [103]. The rational connection to APRO proteins and MMP proteins could be elucidated through the intensive upcoming research in terms of cancer therapeutics.

## 7. Conclusions

Although APRO family proteins could regulate the invasiveness of cancer cells, alternative consequences might occur due to the special effects of MMPs and/or exosomes containing certain microRNAs.

## Figures and Tables

**Figure 1 cancers-14-04931-f001:**
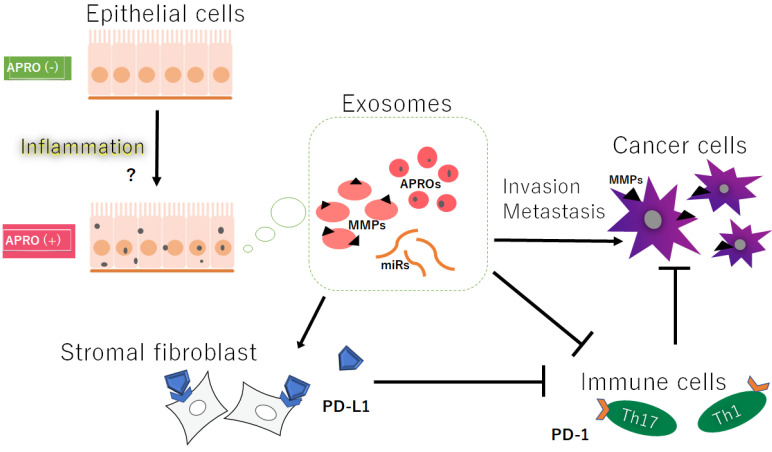
Hypothetical schematic image of the relationship among the APRO proteins (APROs), exosomes, immune cells, immune check point PD-L1 on stromal fibroblasts, and the cancer cells’ invasion/metastasis. Indicated molecules are examples. Arrowhead means stimulation, whereas hammerhead represents inhibition. Note that some critical pathways such as inflammation activation and/or cancer cell growth pathway have been omitted for clarity. Abbreviations: APROs—APRO family proteins; miRNAs—microRNAs; MMPs—matrix metalloproteinases; PD-L1—programmed cell-death ligand 1.

**Figure 2 cancers-14-04931-f002:**
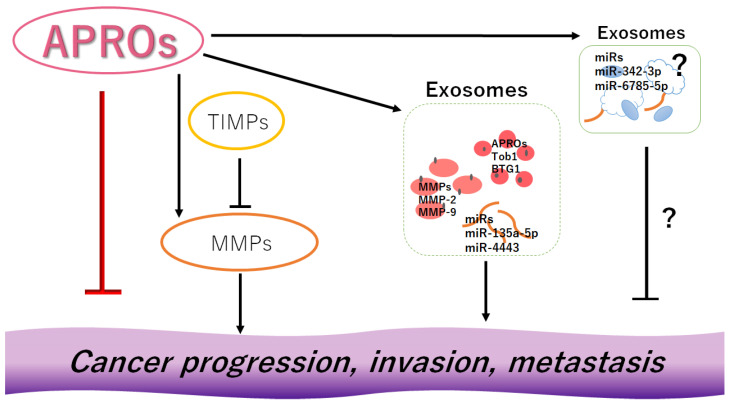
Although APRO family proteins (APROs) could individually inhibit the progression, invasion, and/or metastasis of cancer cells, the other conditions such as the expression levels of MMPs or TIMPs, the presence of some exosomes, and/or the function of microRNAs (miRNAs) or specific proteins in the exosomes could further alter the effect of APRO proteins, either of promotion or inhibition, on the invasiveness of cancer cells. Indicated molecules are examples. Arrowhead means stimulation, whereas hammerhead represents inhibition. Note that some critical pathways have been omitted for clarity. Abbreviations: APROs—APRO family proteins; miRNAs—microRNAs; MMPs—matrix metalloproteinases; TIMPs—tissue inhibitors of metalloproteinases.

## Data Availability

The data presented in this study are available in this article.

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
