# Peer review of "Presumed Roles of APRO Family Proteins in Cancer Invasiveness"

_cancers, 2022, doi:10.3390/cancers14194931_

Round 1
Reviewer 1 Report
This review article attempts to summarize the role of the APRO protein family, MMPs, TIMPs, and miRNAs in cancer progression and invasion.
In itself, the presentation of APRO and the description of its functions, like the similar presentation of MMPs, is basically correct, but some aspects are not clear.
1.The role of MMPs is not conditional but proven in cancer progression and invasion. Many sentences in the chapters on MMP should be rephrased as facts.
2. It is not clear, even at the hypothesis level, what the molecular relationship is between the APRO family and MMPs. The article is a bit like mixing two different reviews together. I would definitely recommend writing a chapter that actually discusses the APRO-MMP interacting effects through examples.
3. There are many typos in the article (e.g.: AORO, ivasion, migratioin, etc.). Their correction is justified.
I suggest major revision.
Author Response
Rev1
Comments and Suggestions for Authors
This review article attempts to summarize the role of the APRO protein family, MMPs, TIMPs, and miRNAs in cancer progression and invasion.
In itself, the presentation of APRO and the description of its functions, like the similar presentation of MMPs, is basically correct, but some aspects are not clear.
- The role of MMPs is not conditional but proven in cancer progression and invasion. Many sentences in the chapters on MMP should be rephrased as facts.
According to this comment, we have rephrased several sentences to the facts we can see.
- It is not clear, even at the hypothesis level, what the molecular relationship is between the APRO family and MMPs. The article is a bit like mixing two different reviews together. I would definitely recommend writing a chapter that actually discusses the APRO-MMP interacting effects through examples.
This is a good point. According to this comment, we would prefer to change the manuscript-type from Review to Perspective. The title has been also replaced with “Assumed roles of APRO family proteins in cancer invasiveness”. In addition, we have repetitively amended the manuscript more helpful to the readers. It might take large space to precisely introduce APRO proteins. Besides, the physiological roles of APRO proteins still remain unknown. We have mentioned it at the end of Section 5, and cited the reference 100 (Ikeda, Y.; Taniguchi, K.; Nagase, N.; Tsuji, A.; Kitagishi, Y.; Matsuda, S. Reactive oxygen species may influence on the crossroads of stemness, senescence, and carcinogenesis in a cell via the roles of APRO family proteins. Explor Med. 2021, 2, 443–454.).
- There are many typos in the article (e.g.: AORO, ivasion, migratioin, etc.). Their correction is justified.
We have gone over the text/abstract and amended typos, misspellings and grammatical errors in the previous manuscript as much as possible.
I suggest major revision.
Reviewer 2 Report
The authors have nicely written the review on current aspects of Cancer Therapy.
The title is confusing with Manuscript. The title and the content of the manuscripts do not match with the statements the authors wants to declare or stat.
The Authors should focus on APRO proteins rather than discussing different pathways and proteins e.g. MMP-9 or MMP family.
Moreover, the manuscript is struggling with siRNA and combination of proteins they wants to declare. Please clarify the role and rewrite all statements as its very confusing.
Include role of Piwi RNA, si RNA and miRNA AND APRO proteins in Cancer therapeutics.
The scope of this article is very good but the information given in this manuscript is struggling with the scientific data.
Please also go through recent articles and cite them to make this topic more useful.
Please rewrite the title, totally confusing.
Author Response
Rev2
Comments and Suggestions for Authors
The authors have nicely written the review on current aspects of Cancer Therapy.
The title is confusing with Manuscript. The title and the content of the manuscripts do not match with the statements the authors wants to declare or stat.
According to this comment, the title has been replaced with “Assumed roles of APRO family proteins in cancer invasiveness”.
The Authors should focus on APRO proteins rather than discussing different pathways and proteins e.g. MMP-9 or MMP family.
It might take large space to precisely introduce APRO proteins. Besides, the physiological role of APRO proteins still remain unknown. We have added reference-100 that might help to know the various roles of APROs.
Moreover, the manuscript is struggling with siRNA and combination of proteins they wants to declare. Please clarify the role and rewrite all statements as its very confusing.
We have gone over the text/abstract and amended typos, misspellings and grammatical errors in the previous manuscript as much as possible in order to improve the manuscript more helpful to the readers.
Include role of Piwi RNA, si RNA and miRNA AND APRO proteins in Cancer therapeutics.
Including the role of Piwi RNA and si RNA in this manuscript might be further confusing, however, which is very interesting to me. Please see the reference 100.
The scope of this article is very good but the information given in this manuscript is struggling with the scientific data.
Please also go through recent articles and cite them to make this topic more useful.
We have repetitively amended the manuscript more helpful to the readers.
Please rewrite the title, totally confusing.
OK, we did it.
Reviewer 3 Report
Consider the importance of the APRO family members involving in the regulation of cell growth, migration, and/or invasion, it is important to summarize their roles in cancer disease. This article is well written and gives a good summary on the contradictory reports for the invasiveness brought by cancer cells in relation to the action of APRO family proteins, on the basis of understanding the function of MMPs and/or exosomes.
Author Response
Rev3
Comments and Suggestions for Authors
Consider the importance of the APRO family members involving in the regulation of cell growth, migration, and/or invasion, it is important to summarize their roles in cancer disease. This article is well written and gives a good summary on the contradictory reports for the invasiveness brought by cancer cells in relation to the action of APRO family proteins, on the basis of understanding the function of MMPs and/or exosomes.
Thank you very much for the good evaluation on the manuscript.
Reviewer 4 Report
In this review by Ilkeda et al entitled “To promote or not to promote cancer invasion and/or metastasis with APRO family proteins, that is the problem for cancer therapy”, the authors provide a brief review on the APRO family and their controversial role in regulating carcinogenesis.
The main piece of work is broken down into sections initially to introduce the APRO members and subsequently to highlight the role of miRNA, exosomes, and MMPs in cancer. The involvement of APRO in the different contexts is not always clear.
A large proportion of the arguments seem to relate to the importance of miRNA or MMPs in carcinogenesis and only very sporadically do APRO protein actually come in the context of this.
MMPs and miRNA have been shown to be key regulators of cancer progression for many years, if not decades, and this review only scratches the surface on this topic, making it insignificant compared to much broader analysis. If on the other hand, the authors wish to combine MMPs and miRNAs in the context of APRO regulation, this must be much more highlighted where the latter is key and the focus of the work.
Author Response
Rev4
Comments and Suggestions for Authors
In this review by Ilkeda et al entitled “To promote or not to promote cancer invasion and/or metastasis with APRO family proteins, that is the problem for cancer therapy”, the authors provide a brief review on the APRO family and their controversial role in regulating carcinogenesis.
The main piece of work is broken down into sections initially to introduce the APRO members and subsequently to highlight the role of miRNA, exosomes, and MMPs in cancer. The involvement of APRO in the different contexts is not always clear.
According to this comment, we would prefer to change the manuscript type from Review to Perspective. In addition, we have repetitively amended the manuscript more helpful to the readers. Again, thank you very much for the kind suggestion.
A large proportion of the arguments seem to relate to the importance of miRNA or MMPs in carcinogenesis and only very sporadically do APRO protein actually come in the context of this.
It might take large space to precisely introduce APRO proteins. Besides, the physiological roles of APRO proteins still remain unknown. We have mentioned it at the end of Section 5, and cited the reference 100 (Ikeda, Y.; Taniguchi, K.; Nagase, N.; Tsuji, A.; Kitagishi, Y.; Matsuda, S. Reactive oxygen species may influence on the crossroads of stemness, senescence, and carcinogenesis in a cell via the roles of APRO family proteins. Explor Med. 2021, 2, 443–454.).
MMPs and miRNA have been shown to be key regulators of cancer progression for many years, if not decades, and this review only scratches the surface on this topic, making it insignificant compared to much broader analysis. If on the other hand, the authors wish to combine MMPs and miRNAs in the context of APRO regulation, this must be much more highlighted where the latter is key and the focus of the work.
This is a very good point. According to this excellent suggestion, we have added the sentence “We wish we could combine MMPs and exosomes in the context of APRO regulation for the invasiveness of cancer cells” in the Introduction section.
Round 2
Reviewer 1 Report
The revised and changed-type manuscript is now acceptable for publication.
Author Response
Thank you very much.
Reviewer 2 Report
Although, Authors have improved the manuscript as per previous suggestions the manuscript is still needs improvements.
Authors have not justified few points as per previous suggestions e.g. to add more keen details about the APRO family rather than stating "100th reference is added".
Kindly add the information about PiWi RNA, siRNA and association of cancers-APRO family to make this manuscript meaningful.
Please describe the rational connection to APRO proteins and MMP proteins in terms of cancer therapeutics.
Author Response
according to this suggestion, we have amended the text, and added some comments and 3 references at the end of Section 6.
Thank you very much for the good suggestion.
Round 3
Reviewer 2 Report
The authors have improved the manuscript as per previous suggestions. I recommend revising the title as the title is not suitable compared to the content of this manuscript.
Author Response
According to this comment, the title had been replaced with “Assumed roles of APRO family proteins in cancer invasiveness”.
Do you think this title is still not suitable?
If so, we would like to change the title as " Presumed roles of APRO family proteins in cancer invasiveness".